# Knowledge Development in Artificial Intelligence Use in Paediatrics

**Peter Kokol** [1,*] , **Helena Blažun Vošner** [2,3,4] **and Jernej Završnik** [2,3,5,6]

1   Faculty of Electrical Engineering and Computer Science, University of Maribor, 2000 Maribor, Slovenia
2   Community Healthcare Centre Dr. Adolf Drolc Maribor, 2000 Maribor, Slovenia; helena.blazun@zd-mb.si (H.B.V.); jernej.zavrsnik@zd-mb.si (J.Z.)
3   Alma Mater Europaea—ECM, 2000 Maribor, Slovenia
4   Faculty of Health and Social Sciences Slovenj Gradec, 2380 Slovenj Gradec, Slovenia
5   Faculty of Natural Sciences and Mathematics, University of Maribor, 2000 Maribor, Slovenia
6   Science and Research Center Koper, 6000 Koper, Slovenia
*   Correspondence: peter.kokol@um.si

**Abstract:** The use of artificial intelligence in paediatrics has vastly increased in the last few years. Interestingly, no historical bibliometric study analysing the knowledge development in this specific paediatric field has been performed yet, thus our study aimed to close this gap. References Publication Years Spectrography (RPYS), more precisely Cited Reference Explorer (CRE) software tool was employed to achieve this aim. We identified 28 influential papers and domain experts validation showed that both, the RPYS method and CRE tool performed adequately in the identification process.

**Keywords:** keyword artificial intelligence; paediatrics; bibliometrics; influential papers; RPYS; CRE

## 1. Introduction

The use of artificial intelligence (AI) in medicine can be traced back to 1963 when Rand Corporation published its memorandum on AI and brain mechanisms [1]. The descriptions of some of the first practical applications were published in 1968. In these applications AI was used to support diagnosing [2], and using decision trees in prognosing drug effects in children [3]. Few years later Shortliffe et al. [4] presented the expert system Mycin, developed in 1971, which was able to identify bacteria causing severe blood infections and to recommend antibiotics. Despite the fact that Mycin outperformed members of the Stanford medical school in the accuracy of diagnosis it was never used in practice due to a legal issue -who do you sue if it gives a wrong diagnosis? In the same year when Mycin was developed Bellanti & Schlegel [5] published a study in which they presented a paediatric AI application supporting the diagnosing of immune deficiency diseases. However, the use of AI in medicine become really popular only in 2016, when the IBM AI platform Watson correctly and spectacularly diagnosed and proposed an effective treatment for a 60-year-old woman's rare form of leukemia [6]. Since then more than 12,000 papers presenting AI application in paediatrics were published covering themes such as use of machine learning in public health, deep learning in image and signal processing, diagnosing, emotion recognition and serious games for health education [7], indicating that AI become a hot research topic also in paediatrics.

Some recent review papers including a bibliometric based review analysed the use of AI in paediatrics in general [8–10] and further ones reviewing the AI use in paediatrics subspecialities such as: radiology [11,12], prematurity [13,14], surgery [15,16] or cardiology [17,18]. Even though studying the history of and developing knowledge in medical fields is widely recognized [19], the above publications failed to present a structured and systematic analysis of how the use of AI in paediatrics has progressed. Namely, understanding the history of knowledge development. enables medical professionals to learn

from past experiences and influence the practice in clinical environments in more efficient ways. The aim of our current study was to close this gap.

Knowledge development and its impact on the society in general can be studied in various ways. The traditional approach is to analyze citation patterns [20]. A more recent approach emerging lately is called Altmetrics. Altmetrics is a set of quantitative and qualitative metrics, which complements citation metrics. Altmetrics scans mentions and discussions in social and research networks such as Mendeley, Twitter, Facebook, mainstream media, research blogs, reference manager bookmarks and similar [21,22]. Despite Altmetrics raising popularity and impact it is argued that Altmetrics is not sufficient to determine the scientific value of the research output and is also more biased (for example double tweeting [23]) than traditional bibliometrics based approaches [24,25]. Additionally, it has also been shown that tweeting and consequently Altmetrics influence citation rates [26], Due to above reasons we decided to employ the traditional knowledge development analysis in our study.

## 2. Materials and Methods

The notion of historical roots as important or influential publications in a specific research area (SRA) was introduced by Robert K. Merton in 1985 [27]. Since than, historical roots and influential papers identification become part of the scientific discourse in scientometrics and bibliometrics research. Simply, the number of citations that SRA publications received seems to be an obvious measure to identify SRA's historical roots. However, SRA publications might be cited (frequently) also by publications outside the SRA, which suggests that such publication may have impact beyond SRA, but not in the SRA in question. Moreover, the SRA's cited publications may not be indexed in bibliographic databases at all. Consequently, a method called References Publication Years Spectroscopy (RPYS) has been developed to overcome this problem. One of the core software tools implementing RPYS is the Cited Reference Explorer (CRE; www.crexplorer.net 10 February 2022) [28,29], which we employed in our study. The method has been already successfully used in different medical SRAs [30,31]. CRE analyses references' publication years and aggregate the number of cited references over time on a spectrogram. Pronounced peaks indicates the years when historical roots/influential papers were published. In addition to the spectrogram CRE offers some other tools to identify influential publications. In our study we used the CRE tabular output to identify historical roots for the early period and the CRE spectrogram for the recent period. Additionally, CRE indicators, namely N_TOP10, which reveals publications that were among the 10% most cited publications over a longer period and Sleeping beauties (SB) indicators were also used. An SB is a publication that goes unnoticed (sleeps) for a long time and then almost suddenly become highly cited and turns out to be interesting (awakens). SBs frequently presented important innovations and their awakening can be associated with important paradigm shifts in science [32,33]. The identified influential publications were reviewed by five domain experts coming from computer and health science fields to informally asses the accuracy of the identification [34].

As part of the analysis, we created a corpus of publications related to the application of AI in paediatrics using the SCOPUS bibliographic database employing the search string ("machine learning" or "rough sets" or ("decision tree*" and (induction or heuristic)) or "artificial neural network*" or "support vector machines" or "rough sets" or "deep learning" or "intelligent systems" or "artificial intelligence") and (newborn* or toddler* or child* or adolescent*) and exported their metadata to the CRE. The search was performed on 3rd of October 2021.

## 3. Results

The search resulted in 12,360 publications citing 977,812 references. After removing duplicates 781,526 references remained and were analysed with the CRE. The analysis resulted in 28 historical roots/influential papers, which are presented in Figure 1 as peaks in the blue line and compiled in Table 1. The oldest cited reference, namely Hobbes

Leviathan [35] was published in 1651. Regarding the content, the references can be divided into three groups:

- Publications concerned with general intelligence, emotions, psychology, behaviour and functioning of brains,
- Publications concerning statistical and mathematical methods, and
- Computer algorithms, tools and languages used for building artificial intelligent systems.

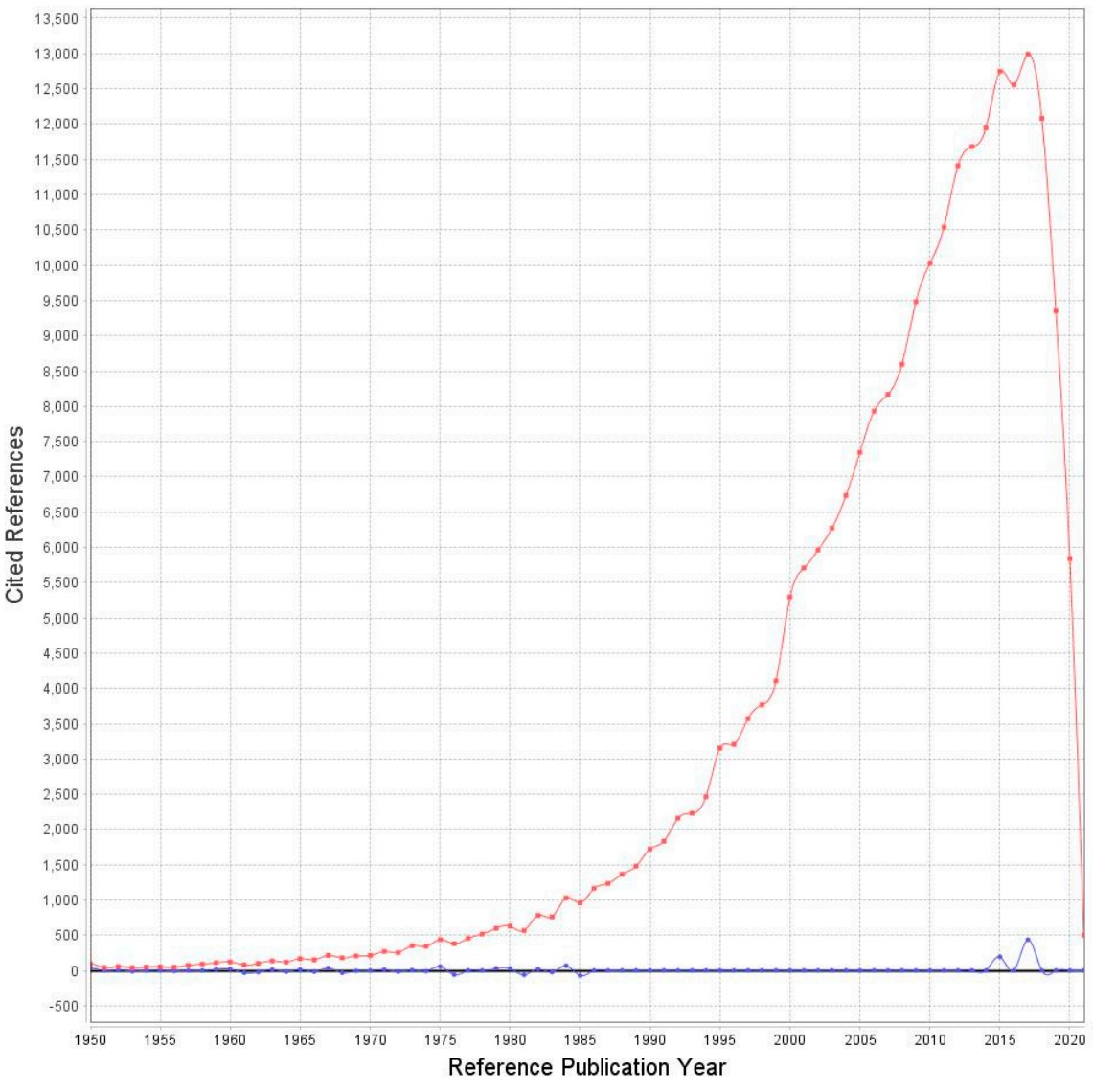

**Figure 1.** Paediatric historical roots in area of using Artificial intelligence in paediatrics for the period 1950 (the year when A. Turings, Computing machinery and intelligence was published) to 2021. Peaks in blue curve represent years when important publications were published.

Knowledge development vent trough several stages, namely (1) researching general intelligence, emotions and attitudes, (2) establishing statistical and mathematical foundations, (3) brain and neural networks research, (4) computing machinery and intelligence (5) development of AI algorithms and their applications in paediatrics, and (6) Deep learning applications in paediatrics.

**Table 1.** Historical roots in the area of using Artificial intelligence in paediatrics.

| Authors | Year | Title | |
| --- | --- | --- | --- |
| Hobbes, T | 1651 | Leviathan | Oldest |
| Darwin, C., | 1872 | The expression of the emotions in man and animals | Tabular |
| James, W., | 1890 | Principles of Psychology | Tabular |
| Pearson, K., | 1901 | On lines and planes of closest fit to systems of points in space | Tabular |
| Spearman, C. | 1904 | General intelligence, "objectively determined and measured" | Tabular |
| Spearman, C. | 1904 | The proof and measurement of association between two things | Tabular |
| Likert, R. | 1932 | A Technique for the Measurement of Attitudes | Tabular |
| Hotelling, H. | 1933 | Analysis of a complex of statistical variables into principal components | Tabular |
| Fisher, R.A | 1936 | The use of multiple measurements in taxonomic problems | Tabular |
| Kanner, L. | 1943 | Autistic disturbances of affective contact | Tabular |
| McCulloch, W.S., Pitts, W | 1943 | A logical calculus of the ideas immanent in nervous activity | Tabular |
| Dice, L.R., | 1945 | Measures of the amount of ecologic association between species | Tabular |
| Hebb, D.O., (1949) | 1949 | The Organization of Behaviour | Tabular |
| Turing, A.M. | 1950 | Computing machinery and intelligence | Spectrogram |
| Zadeh, L.A | 1965 | Fuzzy sets | Spectrogram |
| Cover, T., Hart, P | 1967 | Nearest neighbour pattern classification | Spectrogram |
| Jennett, B., Bond, M | 1975 | Assessment of outcome after severe brain damage | Spectrogram |
| Breiman, L., Friedman, J., Stone, C.J., Olshen, R.A. | 1984 | Classification and Regression Trees | Spectrogram |
| Quinlan, J.R. | 1986 | Induction of decision trees | NTop10 |
| Rumelhart, D.E., Hinton, G.E., Williams, R.J. | 1986 | Learning representations by back-propagating errors | NTop10 |
| Quinlan, J.R. | 1993 | C4.5: Programs for Machine Learning | NTop10 |
| Bishop, C.M | 1995 | Neural Networks for Pattern Recognition | NTop10 |
| Pedregosa, F. | 2011 | Scikit-learn: Machine learning in Python | SB |
| Krizhevsky, A., Sutskever, I., Hinton, G.E. | 2012 | Imagenet classification with deep convolutional neural networks | SB |
| LeCun, Y., Bengio, Y. | 2015 | Deep learning | Spectrogram |
| Esteva, A., Kuprel, B., Novoa, R.A. | 2017 | Dermatologist-level classification of skin cancer with deep neural networks | Spectrogram |

## 4. Discussion

During the expert review it was concluded that all of the identified influential publications might have contributed to the development of Artificial intelligence in general and its use in paediatrics, however some interesting observations have been made, which could lead to further investigations:

- Robotic surgery is quite routinely used in medicine, and "emotion teaching" or "stress reduction robots" are widely used in paediatrics, but none influential papers related to robotics or history of robotics were identified. It could be the case that earliest papers on robotics were not scientific papers, but less known technical reports. Earlier mentions of Greek, Persian and Medieval automata are also not documented (or that documentation has been destroyed or lost) and become known only recently in various histories of AI.
- Statistical and mathematical papers did not directly contribute to the development of AI, however advanced statistics and mathematics are used in proving of AI algorithms, especially in machine learning.
- There are a lot of influential papers related to prominent machine learning algorithms, such as neural networks, deep learning, decision trees and fuzzy sets, and also in paediatrics rarely used algorithms like nearest neighbors, however influential publications on popular algorithms such as support vector machines or Bayes are missing.

## 5. Conclusions

In terms of methodologically, RPYS supported by CRE proved to be a viable approach to identify historical roots or influential papers. Addtionaly. experts also agreed that the analysis of knowledge development based on historical roots identification can significantly contribute to the understanding specific research fields, provide learning from past experiences, or provide motivation for further research and consequently enable health care professionals to influence the everyday practice settings using historical evidence.

**Author Contributions:** Conceptualization, P.K. and J.Z.; methodology, P.K.; validation, P.K., J.Z. and H.B.V.; formal analysis, P.K.; investigation, H.B.V.; resources, J.Z.; data curation, P.K.; writing—original draft preparation, P.K.; writing—review and editing, J.Z., H.B.V.; visualization, H.B.V.; supervision, J.Z. All authors have read and agreed to the published version of the manuscript.

**Funding:** This research received no external funding.

**Institutional Review Board Statement:** Not applicable.

**Informed Consent Statement:** Not applicable.

**Data Availability Statement:** Not applicable.

**Conflicts of Interest:** The authors declare no conflict of interest.

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
