# Peer review of "Knowledge Development in Artificial Intelligence Use in Paediatrics"

_knowledge, doi:10.3390/knowledge2020011_

Round 1

Reviewer 1 Report

The review of the solutions used so far is a very important element of the research work. The above-mentioned method with the use of RPYS and CRE tools allows to accelerate and automate the process of research resarch. However, as the authors themselves note in Section 3, the method did not prove to be fully effective, both due to the limited scope of cheese strings and due to the fact that some works important from a research perspective are not typical scientific articles, but technical reports.

In addition, I would like to point out that citation is an important, but not the only criterion showing the impact of scientific work on the development of a given field. In recent years, many magazines also disclose statistics on the number of views and downloads of a given article, which also proves the impact of a given work on the scientific community.

I believe that "Knowledge Development in Artificial Intelligence Use in Paediatrics" as a brief review should be extended to include this aspect.

Author Response

We would like to thank the reviewer for the valuable comments. We extended the introduction with a short description of altmetrics  and the reasons why we selected the traditional knowledge development analysis approach in our study. We also proofread and edit the paper

Reviewer 2 Report

In this paper, the authors argue for a survey on the use of artificial intelligence in the world of paediatrics medicine.
The authors argue that there are no studies on the bibliography related to the field of examination.
In order to produce a comprehensive study, a spectrography of publication years is compiled using the CitedReferenceExplore software tool. 
From the proposed graph, it appears that the healthy authors were able to identify influential articles and validation from experts in the field. In fact, they showed that both the RPYS method and the CRE tool worked adequately in the identification process. 

strengths:
I think this work is very interesting and aims to fill an equally important gap that the authors should better define.

weaknesses:
The authors while very good at identifying gaps were not good at defining them well. In fact, already in the introduction, they should be more pronounced.

One struggles to understand the point of your work.

I recommend reviewing from line 46-50.
I also recommend splitting the Results section from Discussions for better readability.

Author Response

Thank you for your valuable comments. The second part of the introduction has been rewritten to make the research gap more explicit. The results and discussion has been split into two chapters. Paper has been proofread.

Reviewer 3 Report

The authors provide an interesting study on the use of RPYS supported by CRE to identify influential manuscripts using AI in pediatrics. While the study is important and can be accepted, a minor concern is the use of terminology “References Publication Years Spectrography (RPYS)” over “References Publication Years Spectroscopy (RPYS)”. The authors should comment on it.

Author Response

Thank you for your valuable comments. We changed spectrography to spectroscopy and proofread the paper

Round 2

Reviewer 1 Report

Thank you for completing the Introduction section with information about Altmetrics. In this way, the authors indicated the existence of other criteria for assessing the impact of scientific work on the development of scientific communities and explained why they did not include them in their work. In its current form, "Development of knowledge on the use of artificial intelligence in pediatrics" fully meets the criteria of a brief review.